# Effect of non-pharmaceutical interventions in the early phase of the COVID-19 epidemic in Saudi Arabia

**Javier Perez-Saez**[1], **Elizabeth C. Lee**[1], **Nikolas I. Wada**[2], **Ada Mohammed Alqunaibet**[3], **Sami Saeed Almudarra**[4], **Reem F. Alsukait**[5,6], **Di Dong**[6], **Yi Zhang**[6], **Sameh El Saharty**[6], **Christopher H. Herbst**[6], **Justin Lessler**[1,7] *

1 Department of Epidemiology, Johns Hopkins Bloomberg School of Public Health, Baltimore, MD, United States of America, 2 Johns Hopkins Novel Coronavirus Research Compendium, Johns Hopkins Bloomberg School of Public Health, Baltimore, MD, United States of America, 3 Public Health Authority, Riyadh, Saudi Arabia, 4 Ministry of Health, Riyadh, Saudi Arabia, 5 Community Health Sciences, College of Applied Medical Sciences, King Saud University, Riyadh, Saudi Arabia, 6 Health, Nutrition and Population Global Practice, World Bank Group, Washington DC, United States of America, 7 Department of Epidemiology, Gillings School of Global Public Health, University of North Carolina, Chapel Hill, NC, United States of America

* jlessler@unc.edu

**Data Availability Statement:** Data has been provided as supplementary material.

## Abstract

Non-pharmaceutical interventions have been widely employed to control the COVID-19 pandemic. Their associated effect on SARS-CoV-2 transmission have however been unequally studied across regions. Few studies have focused on the Gulf states despite their potential role for global pandemic spread, in particular in the Kingdom of Saudi Arabia through religious pilgrimages. We study the association between NPIs and SARS-CoV-2 transmission in the Kingdom of Saudi Arabia during the first pandemic wave between March and October 2020. We infer associations between NPIs introduction and lifting through a spatial SEIR-type model that allows for inferences of region-specific changes in transmission intensity. We find that reductions in transmission were associated with NPIs implemented shortly after the first reported case including Isolate and Test with School Closure (region-level mean estimates of the reduction in $R_0$ ranged from 25–41%), Curfew (20–70% reduction), and Lockdown (50–60% reduction), although uncertainty in the estimates was high, particularly for the Isolate and Test with School Closure NPI (95% Credible Intervals from 1% to 73% across regions). Transmission was found to increase progressively in most regions during the last part of NPI relaxation phases. These results can help informing the policy makers in the planning of NPI scenarios as the pandemic evolves with the emergence of SARS-CoV-2 variants and the availability of vaccination.

## Introduction

Community SARS-CoV-2 transmission was first reported in the Middle Eastern region in early to late January 2020 in United Arab Emirates, although the extent of undocumented

**Funding:** JPS, ECE, RE, DI, YZ, CHH and JL were supported by the Saudi Ministry of Finance under the Health, Nutrition and Population Reimbursable Advisory Services Program (P172148) between the World Bank and the Saudi Public Health Authority. The funders had no role in study design, data collection and analysis, decision to publish, or preparation of the manuscript.

**Competing interests:** No competing interests to declare

transmission remains unclear due to important differences in testing capacity [1]. The first imported COVID-19 case in Saudi Arabia was reported on March 2nd, 2020 [2], and by October 10, 2020, more than 300,000 confirmed cases and 4,923 deaths had been reported (Fig 1). The intensity of the Saudi Arabian epidemic has varied by region, with cumulative reported cases ranging from 20 to 140 per 10,000 people, and deaths from less than 0.5 to more than 2 per 10,000 people (Fig 1).

During the initial phase of the pandemic, the primary tools employed by national and regional governments to interrupt transmission were non-pharmaceutical interventions (NPIs), a broad spectrum of public health measures ranging from informational campaigns to business closures and stay at home orders [3–6]. Saudi Arabia's experience with the ongoing Middle East Respiratory Syndrome Coronavirus (MERS-CoV) outbreak (in which Saudi Arabia has tallied the majority of cases since 2012) may have prompted a more rapid response than would have been implemented otherwise. The Saudi government began implementing travel advisories and restrictions on February 19, 2020. In the two weeks following the first reported case in Saudi Arabia, the Ministry of Health and other Saudi government agencies implemented a series of NPIs, including the suspension of the Umrah religious pilgrimage that draws millions of visitors each year [7], school closures, mandatory quarantine for workers entering the country, and the closure of restaurants and leisure venues [8].

The intensity of NPIs increased as case counts continued to rise. Prayers at mosques were suspended on March 17, domestic flights and all public transportation were suspended on March 21, and a partial curfew was imposed for three weeks beginning on March 23. NPI implementation culminated in a national 5-day lockdown corresponding with the Eid al-Fitr holiday from May 23–27. The first phase of the easing of restrictions began on May 28; the second phase began on May 31, and the final phase began on June 21 with the resumption of

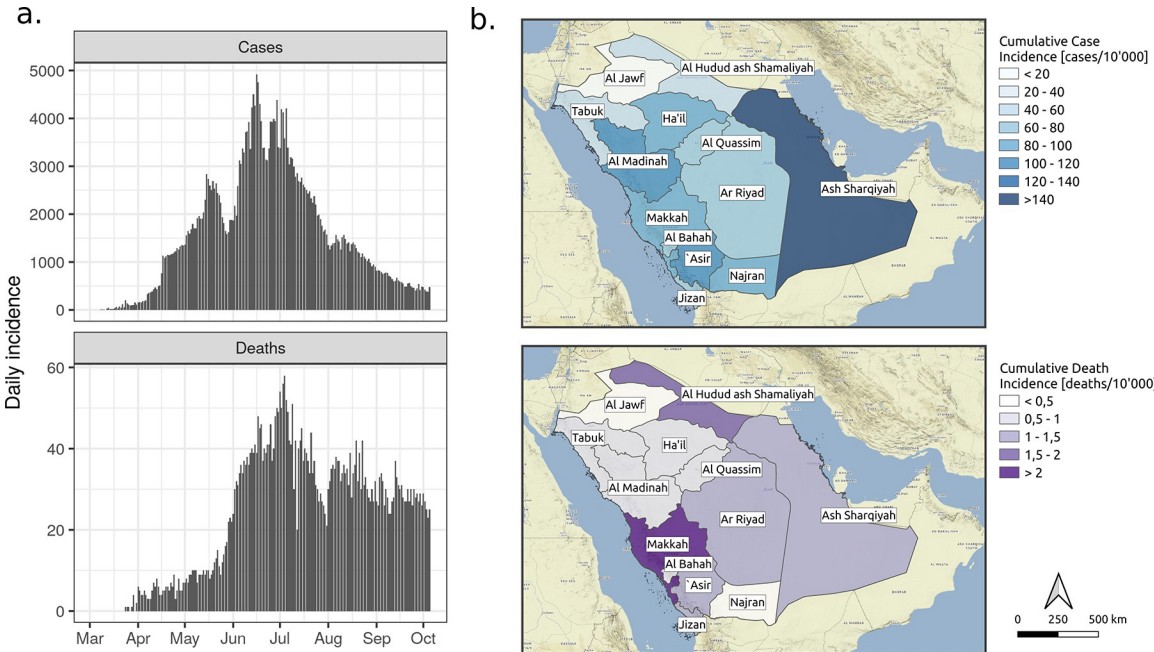

**Fig 1. COVID-19 epidemiological data.** a) National-level daily confirmed COVID-19 case and death incidence from March 2 to October 10 2020. b) Maps of regional cumulative case and death incidence rates per 10,000 population as of October 10 2020. Base layer is Stamen terrain pulled from http://maps.stamen.com (CC BY 3.0.).

most economic activities. However, as of November 6, 2020, schools remained closed and international travel was limited.

Arab states of the Persian Gulf, including Saudi Arabia, have a distinct demographic profile relative to other regions, with a high proportion of the population represented by foreign nationals (immigrants represented 38% of the population of Saudi Arabia in 2019 [CIA Factbook]). Moreover, Saudi Arabia is annually visited by a large number of religious pilgrims from outside the country, for Umrah throughout the year and for the Hajj during a two-week period, though these pilgrimages were substantially limited in 2020 [9]. Despite these distinctive features of the population, only a few published studies have focused on SARS-CoV-2 dynamics in Gulf states [10,11], or specifically in Saudi Arabia [12,13].

Similarly, few studies aimed at estimating NPI effectiveness have focused on the Gulf states. One study simulated the effect of lockdowns in Kuwait, concluding that the ensuing increased household contact rates would lead to greater SARS-CoV-2 transmission among non-Kuwaiti workers of lower socioeconomic status [14]. Some studies aimed at estimating global NPI effectiveness have omitted the Gulf states entirely (e.g., [4,15]). Saudi Arabia has been included in some global NPI effectiveness studies that aggregate data from multiple countries (e.g., [16,17]), and the timeline of interventions in Saudi Arabia has been chronicled [8,18,19]. However, to our knowledge, there have been a few studies that empirically assess NPI effectiveness for Saudi Arabia in particular.

In this study, we estimate the association between NPI implementation and changes in SARS-CoV-2 transmission in Saudi Arabia that were implemented from March to October 2020. We fit a spatial SEIR-type model of SARS-CoV-2 transmission to observed regional-level data that explicitly accounts for changes in transmission intensity according to the timeline of NPI measures implemented in Saudi Arabia.

## Methods

### Epidemiological data and non-pharmaceutical interventions

We obtained region-level and national-level data on daily incident cases and deaths from the Ministry of Health of the Kingdom of Saudi Arabia COVID-19 dashboard API (https://covid19.moh.gov.sa/). Model calibration was based on data reported from March 2, 2020 through October 10, 2020.

Region-level timelines on the policy implementation of NPIs in the Kingdom were compiled from news and government websites such as Ministry of Health, Alarabiyah, Alriyadh, Arab News, Saudi Gazette, Saudi Press Agency, and CNN Arabic, as well as government reports (in Arabic). We identified the following NPIs (Fig 2):

- *International travel ban*: All land borders were shut on March 8, and international flights were suspended on March 14.

- *School Closure*: Schools of all grades were suspended on March 8, and this action continued throughout the study period.

- *Isolate and Test*: This policy began on March 8, with advice to self-isolate when arriving from Italy, South Korea, Egypt and Lebanon. As of March 13, all workers who entered the country were ordered to stay at home and self-quarantine. Targeted mass testing in high-risk areas began on April 17.

- *Public and private gatherings*: On March 15, malls were closed, and restaurants and cafes were barred from serving food on site (take-out orders were allowed). Most government agencies halted in-person workplace attendance. Prayers in all mosques were suspended on

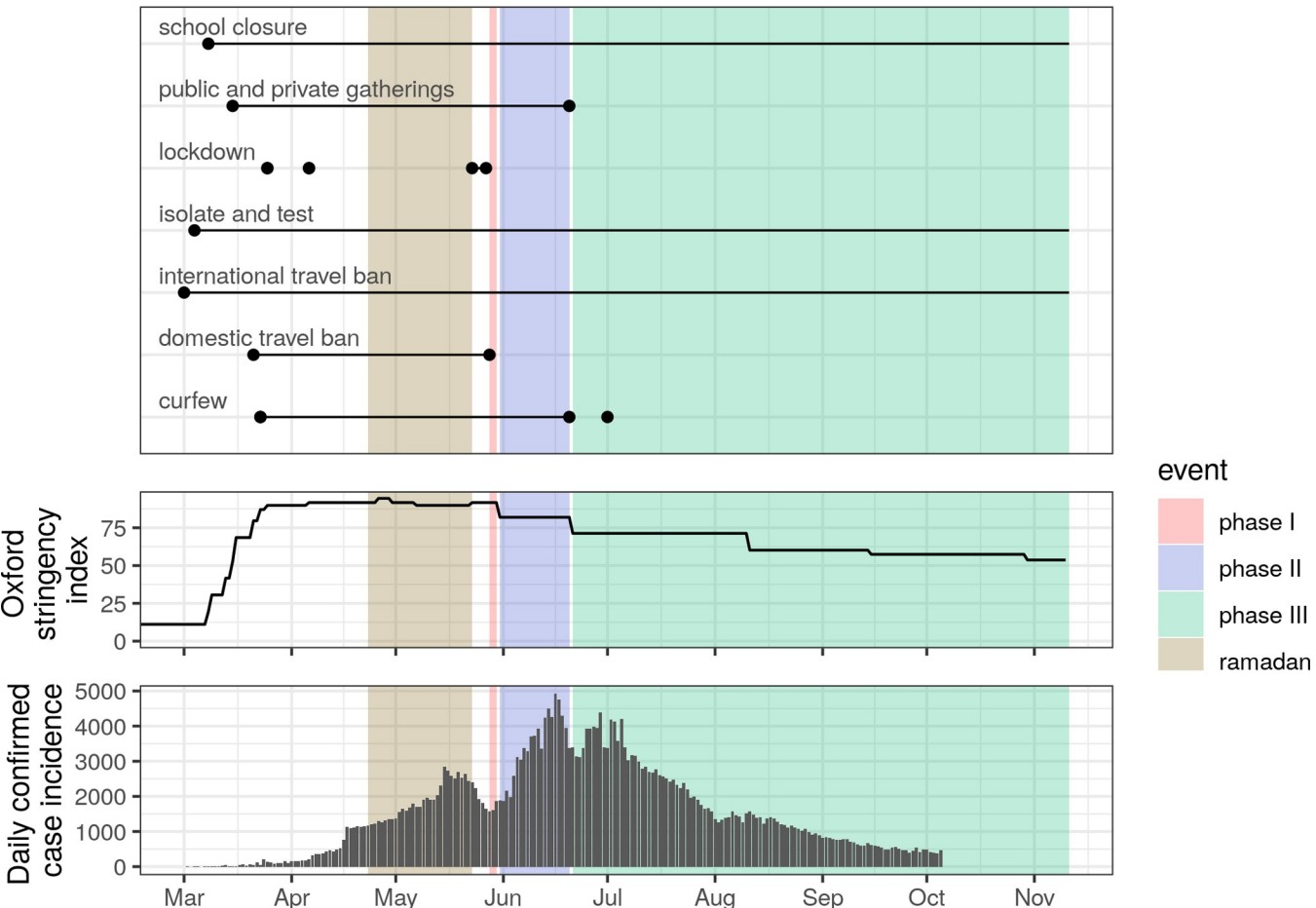

**Fig 2. Dates of NPI implementation in response to COVID-19.** Top: Dates of NPI implementation and changes to status. Circles represent NPI start and end dates, and open lines NPIs that were still in effect on November 11 2020. Phases I, II and III correspond to the phases of NPI relaxation. Middle: The Oxford NPI stringency index provides an external quantitative metric of the strictness of government policies. Bottom: Daily confirmed case incidence across Saudi Arabia.

March 17, except for the Two Holy Mosques in which capacity was reduce capacity and physical distancing measures were implemented.

- *Domestic travel ban*: Domestic flights, buses, and taxis were suspended on March 20.

- *Curfew—part one*: Curfew was declared on March 23 for 21 days between 7 p.m. and 6 a.m.

- *Curfew—part two*: Curfew was declared for the Ramadan period starting on April 26th, limiting movement between 5 p.m. and 9 a.m.

- *Lockdown*: A 24-hour curfew was imposed during the five-day Eid Al-Fitr holiday from May 23 to May 27.

- *Phase I*: The first phase of NPI relaxation occurred from May 28 to May 30. Curfew was eased between the hours of 6 a.m. to 3 p.m. Domestic travel was allowed in private vehicles. Malls and retail shops were allowed to open. Areas where physical distancing could not be enforced remained closed (e.g., beauty salons, sports and health clubs, and cinemas).

- *Phase II*: The second phase of NPI relaxation occurred from May 31 to June 20. Curfew hours were further eased from 6 a.m. to 8 p.m. Domestic flights and inter-region travel were allowed. Restaurants and cafes were allowed to re-open. Group prayers were allowed in mosques.

- *Phase III*: On June 21, curfew was lifted, and all activities were re-opened with the exceptions of the Umrah pilgrimage, international flights, and schools.

For further exploration of NPI effectiveness, we compared the timeline of NPI implementation with an external data source, the Oxford COVID-19 Government Response Stringency index (Hale et al. 2020), a quantitative measure of the strictness of government policies regulating population behavior (Fig 2).

## COVID-19 transmission model

We estimated NPI effectiveness through a spatial SEIR-type model of COVID-19 transmission developed for the purpose of scenario planning and comparison, as described elsewhere [20]. Briefly, the model simulates infection dynamics between a set of connected spatial locations based on parameter estimates for the natural history of SARS-CoV-2. The model consists of a compartmental SEIR model with three infected compartments representing transmission dynamics in each distinct region, with transmission occurring between locations as well. In this analysis we did not consider age stratification. In the case of Saudi Arabia, we modeled transmission dynamics at the administrative 1 (region) level (Fig 1). Population-level mobility fluxes between regions were obtained from Facebook Data for Good, which represents the movement of Facebook users on mobile devices that provide GPS location data. NPIs were implemented as region-specific reductions to the baseline basic reproduction number, the number of secondary infections caused by an infectee in a completely susceptible population without any interventions. Health outcomes were modeled based on the SEIR simulations by incorporating demographic information as described in [20]. Hospitalization and death probabilities were adjusted to site-specific demographics from Worldpop [21] based on published estimates of age-specific health outcomes [22]. The number and timing of epidemic seeding events were estimated in each region by the model, with seeding being allowed in all regions. If not specified otherwise, all model parameters are constant in time, and given in S1 Table.

## Inference on NPI effects

We build an inference framework around our spatial SEIR-type model that fits model outcomes such as confirmed COVID-19 cases and deaths to observed epidemic trajectories. This modeling framework was used to estimate region-specific parameters for: epidemic seeding dates and values, the basic reproduction number, and the effectiveness of different types of phases of NPI implementation. Fixed parameters include the infection fatality ratios, the incubation time, the serial interval, and the delays between infection and case confirmation and death, as detailed in S1 Table. From the estimates of the basic reproduction number and NPI effectiveness, we also estimated the effective reproductive number, which represents the effective number of secondary infections caused by an infectee in a given location and transmission context. Inference on model parameters for each spatial location were drawn jointly given the spatial coupling of COVID-19 transmission dynamics. The framework allows for the use of aggregates of the observation and simulated variables over time periods longer than the ones used for simulation. In this report we jointly calibrate the model to the observed weekly sums of confirmed cases and confirmed deaths at region-levels. We use a Poisson likelihood on the

weekly number of reported cases and a square-root normal likelihood on the number of deaths as detailed in the S1 Text. Details on the inference framework are provided in supplementary text 1.

## Code availability

The COVID-19 modeling pipeline used for this analysis is publicly available on Github at https://github.com/HopkinsIDD/COVIDScenarioPipeline. Epidemiological data used in the analysis are available in S1 File.

## Results

Inferred epidemic trajectories were well calibrated at both regional-level and national-level epidemiological data (Fig 3). Modeled case and death trajectories matched observations in terms of the 95% credible intervals (CrIs), with91% (Al Jawf) to 99.5% (Asir) coverage for incident cases and 92% (Al Madinah) to 100% (Al bahah, Al Hudud Ash Shamaliyah, Hail, Jizan, Najran) of incident deaths,. Modeled estimates for cumulative cases were systematically under-estimated, likely due to poor calibration of the larger national epidemic peaks in May and June (Fig 3).

### Basic reproductive number and NPI effectiveness

Our goal was to estimate the basic reproductive number and the effectiveness of previously implemented NPIs in each region. Due to the overlapping timing of NPIs, it was not possible to estimate the independent effects of specific policies. To improve estimation performance, we performed inference on NPI effectiveness for 7 interventions and periods, grouped as follows: *1) Isolate and Test with School Closure*, *2) Curfew part one*, *3) Curfew part two*, *4) Lockdown*, *5) Phase I relaxation*, *6) Phase II relaxation*, and *7) Phase III relaxation*. The grouping was based on the relative timing of interventions. Given the length of Phase III and potential changes in adherence due to fatigue, we further subdivided this last intervention into two periods for which effectiveness was inferred separately.

Region-level estimates of the basic reproduction number for SARS-CoV-2 had mean estimates in between 1.7 (Al Jawf) and 2.3 (Hail) (Fig 4). Estimates of the effectiveness of combinations of NPI varied significantly between the intervention periods, as well as for the same combination of NPIs between regions. The *Isolate and Test with School Closure* NPI was inferred to have effectiveness between 25% (Makkah, Interquartile Range Credible Intervals (IQR CrIs): 9%-35%) and 41% (Asir, IQR CrIs: 22%-60%), with a mean effect across regions of 36%. Uncertainty in these estimates was considerable, with the 95% CrIs including transmission reductions as low as 1% and as high as 73% across regions. Transmission reductions during both *Curfew* NPIs was heterogeneous among regions, with reductions between 22% (Makkah, IQR CrIs: 9%-34%) and 70% (Najran, IQR CrIs: 64%-84%) during the first half, and 30% (Najran, IQR CrIs: 9%-47%) to 65% (Al Madinah, IQR CrIs: 54%-81%) during the second. Transmission reduction was stronger during *Curfew part two* for all regions except for Najran, Hail, Asir and Al Bahah, with a mean increase in effectiveness of 15%. During the *Lockdown*, which was implemented as a 5-day 24h curfew, mean transmission reductions were between 50% and 60% for all regions, although the uncertainty of the estimates was large (IQR CrIs: 26%-81%).

Turning to relaxation Phases I-III, uncertainty for *Phase I* (May 28 to May 30) was also large, with estimated mean transmission reductions between 35% and 40% for all regions except for Makkah (estimated reduction of 55%, IQR CrIs: 42%-70%), for which remained in Phase I until June 20th. Transmission reduction during the second NPI relaxation phase also presented heterogeneity among regions, with values between 20% (Hailh, IQR CrIs: 8%-29%) and 50% (Makkah, IQR CrIs: 39%-65%). Transmission reduction during both parts of Phase

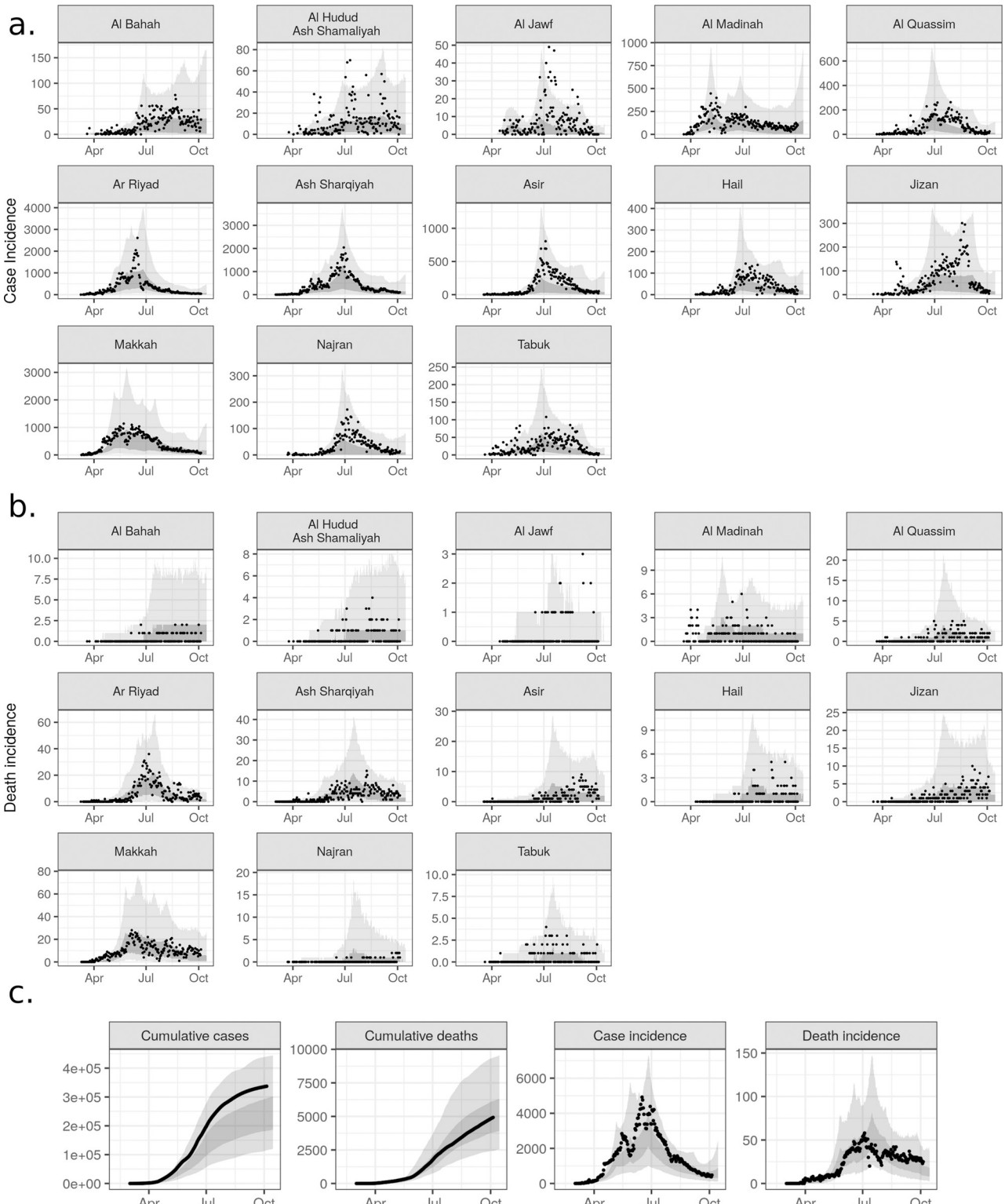

**Fig 3. Model fit to regional-level epidemiological data.** Inferred epidemic trajectories by region. Daily case (a) and death (b) incidence (points) against 95% credible interval (CrI) (light gray ribbons) and 50% CrI (dark gray ribbons) of model trajectory posteriors. c) National-level fits.

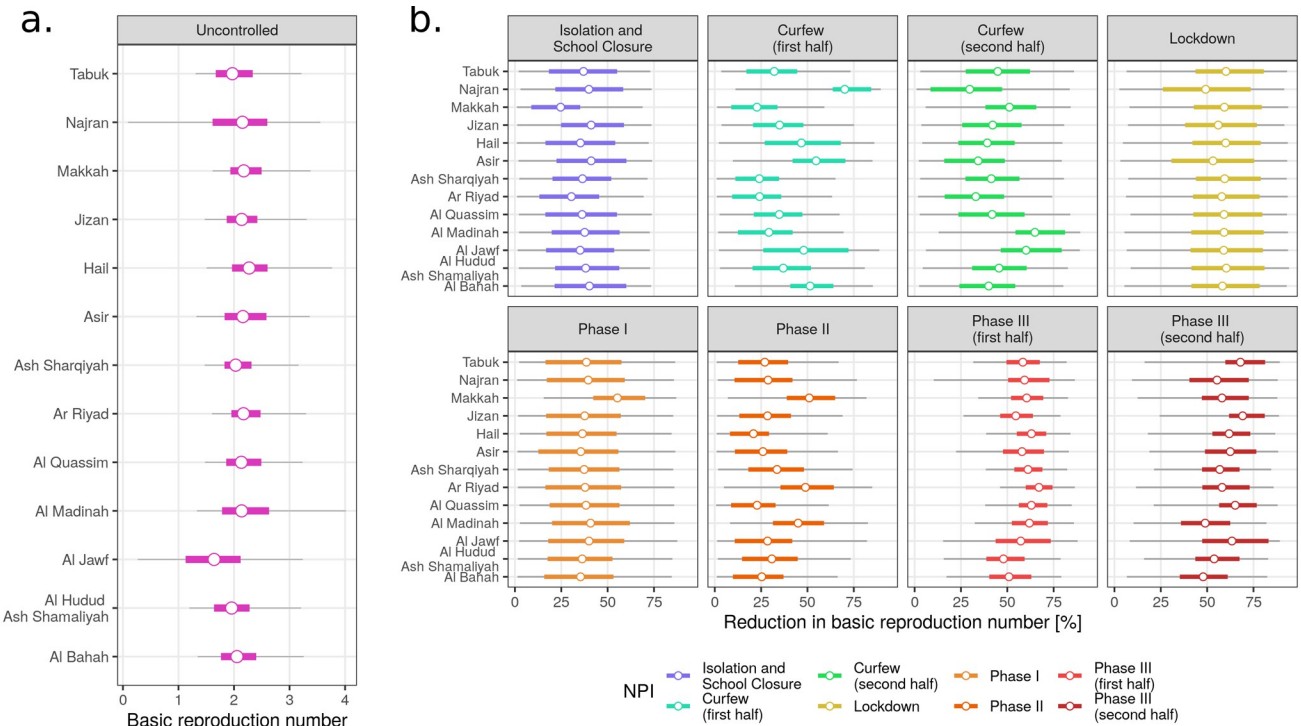

**Fig 4. Inference of baseline R and NPI effectiveness.** a) Inferred basic reproduction number (uncontrolled transmission) by region in terms of the mean posterior estimate (points), and the 95% (black bars) and 50% (red bars) Credible Interval. b) Effectiveness of NPI expressed as the percent reduction of the region-specific basic reproduction number, as shown in the left panel.

III was consistently stronger than that in Phases I and II, with values between 47% (Al Hudud Ash Shamaliyah, IQR CrIs: 39%-59%) and 68% (Ar Riyad, IQR CrIs: 60%-75%) in the first part of Phase III, and between 48% (Al Bahah, IQR CrIs: 35%-61%) and 69% (Jizan, IQR CrIs: 62%-81%) in the second. Transmission reduction tended to weaken during the second half of Phase III in 7 of 13 regions, with a mean transmission increase of 5% across regions.

Varying effectiveness of NPIs implies temporal changes in the effective reproduction number (Fig 5). Transmission reductions during the successive implementation of NPIs resulted in drops of $R_{eff}$ across regions, with mean values below 1 in 11 of 13 regions, although with significant uncertainty. Values of $R_{eff}$ were estimated to increase above 1 during the first NPI relaxation phases in all regions, then to return below 1 in the first half of Phase III. For the second half of Phase III, $R_{eff}$ was estimated to remain near 1 in all regions.

## Discussion

Our analysis found that uncontrolled SARS-CoV-2 transmission in KSA was similar in intensity to that observed in other countries, with an estimated basic reproductive number close to 2 across regions. The NPIs associated with the greatest reductions in transmission were *Isolate and Test with School Closure* (region-level mean estimates of the reduction in $R_0$ ranged from 25 to 41%), *Curfew* (20 to 70% reductions), and *Lockdown* (50 to 60% reductions). However, our estimates of effectiveness were subject to considerable uncertainty, particularly for *Isolate and Test with School Closure* (95% credible intervals from 1% to 73% across regions). Transmission was estimated to have increased progressively during the last phase of NPI relaxation, with the effective reproductive number close to the critical threshold ($R_{eff} = 1$) for most regions at the end of the study period on October 10, 2020.

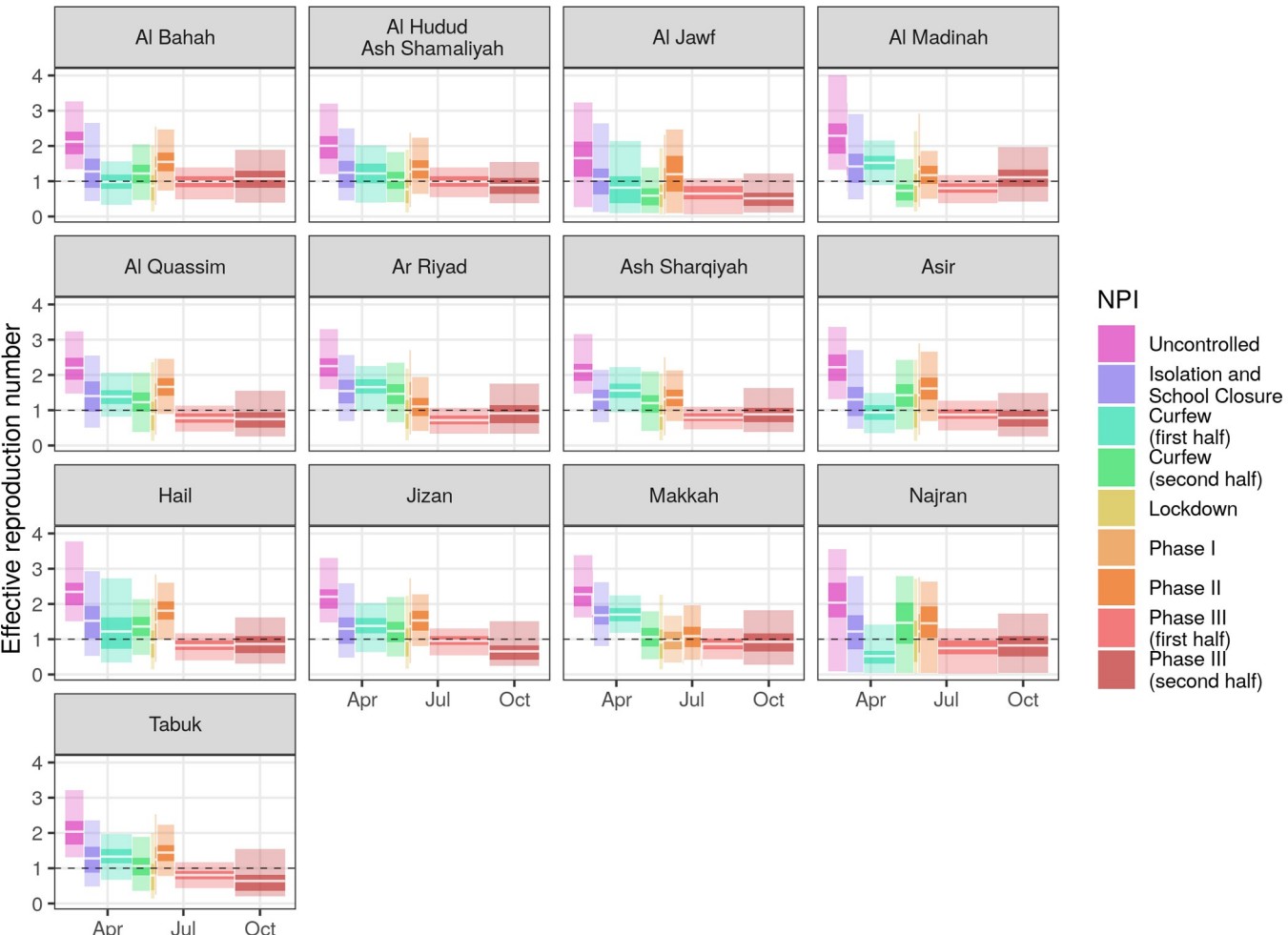

**Fig 5. Changes in the effective reproduction number due to non-pharmaceutical interventions.** Estimates are shown in terms of the median (white lines) and the 50% (dark ribbons) and 95% (light ribbons) CrIs.

These results are in line with evidence from large-scale NPI studies that have found school closures in the spring of 2020 to be associated with SARS-CoV-2 transmission reductions [16,17,23–27]. However, for other interventions, studies attempting to estimate NPI effectiveness have often produced inconsistent results. For example, although some studies found lockdowns to have been effective [3,17,25,28], others found that they were either ineffective on top of other interventions [16,26,29], or could not distinguish their effects from other NPIs [30]. Similarly, the estimated effectiveness of test/trace/isolate programs has varied significantly across studies [16,17,25,31]. There are some exceptions: for example, both theoretical and empirical results consistently indicate that travel restrictions have been useful at slowing epidemic growth at the very beginning of a country's epidemic, but are not particularly effective after local transmission has become established [32–34]. However, for the majority of NPIs, there is the substantial uncertainty in the extent of their practical impact in any given setting, even when there is broad consensus some effect exists.

One key difficulty in all of these approaches is that multiple NPIs were implemented simultaneously or in close temporal proximity to one another; this lack of identifiability makes it extremely difficult to attribute changes in SARS-CoV-2 transmission to any particular

intervention. Moreover, interventions may have complex interactions when implemented at the same time or in succession. KSA is no different in this regard, hence we were compelled to aggregate some disparate interventions into composites (e.g., *Isolate and Test with School Closure*) dictated by their timing. It was particularly difficult to estimate the effect of school closure on changes in transmission, as this NPI was implemented very early (on March 8, six days after the first reported case) and continued throughout the study period. The uncertainty in our effect estimates partially reflects the identification problem inherent in the timeline of Saudi Arabian NPI implementation.

Our analysis was subject to some additional limitations. First, mobility fluxes between regions were assumed to remain constant throughout the simulation period, but because movement limitations were in place, these data may not be representative of current and future inter-regional movement. Second, in the absence of independent infection fatality ratio estimates for Saudi Arabia, we used age-adjusted estimates from other sources, which can have an impact on the case-to-infection ratio and thus on simulated transmission dynamics. Seroprevalence estimates published after our modeling analysis suggest that the infection fatality ratios in regions of KSA may have been lower than the ones we used in this analysis [35], or that there was a systematic under-reporting of deaths, although these seroprevalence estimates were based on convenience samples not representative of the broader population (S1 Text). Third, we did not consider how other types of behavior, such as those influenced by the information environment or by weather, may explain changes in transmission beyond the direct effect of NPIs. Moreover our model assumes homogeneous mixing of people with each region which may be a strong assumption in this specific analysis. Finally, we did not incorporate changes to testing rates, death reporting rates, or the IFR of COVID-19 over time, which means that subsequent inference may not accurately represent transmission dynamics across different phases of the pandemic.

This study also has several key strengths relative to other studies focusing on the Gulf region. First, we account for spatial difference in transmission, NPI efficiency as well as connectivity effects through a spatial SEIR-type model. Second, we exploit both case and death data in a single framework to infer changes in transmission allowing for uncertainty in case reporting. Finally, the same modeling framework we used for inference can be employed to perform simulations of intervention scenarios based on estimates of NPI efficiency. During the study period, three important potential confounding factors were not present: vaccination coverage, the emergence of SARS-Cov-2 variants, and large numbers of foreign visitors for religious pilgrimage (although the Hajj has again been considerably restricted for 2021). As such, these estimates may provide a useful baseline of NPI effectiveness estimates for planning purposes, in the context of several uncertainties related to vaccination coverage, the emergence of SARS-CoV-2 variants, and the Hajj in 2021.

This study is one of the first to estimate transmission dynamics and estimate NPI effectiveness for the Gulf state region in general, and for Saudi Arabia in particular. Many aspects of the COVID-19 epidemic in Saudi Arabia have changed since the study period, and inferences derived in this study may not apply to the future effectiveness of NPIs in the region. Nonetheless, this work stands as a record of what occurred in Saudi Arabia during 2020, and may provide some insight into the comparative effectiveness of NPIs as implemented by the Saudi government.

## Supporting information

**S1 Table. Model parameter values.** Description of fixed and inferred parameters.
(DOCX)

**S2 Table. NPI efficiency estimates.** Estimates are given in terms of the mean and the 95% CrI.
(DOCX)

**S3 Table. Region-specific estimated confirmation probabilities.** Estimates given in terms of
the mean and the 95% CrI.
(DOCX)

**S4 Table. Region-specific estimated seeding dates and amounts.** Estimates given in terms of
the mean and the 95% CrI.
(DOCX)

**S5 Table. Region-specific infection fatality ratios.** IFRs were calculated based on demographic characteristics of each region using the COVIDSeverity R package.
(DOCX)

**S1 Text. Inference algorithm.** Description of the MCMC algorithm used to infer model
parameters.
(DOCX)

**S1 Data. COVID-19 case and death data.** Regional-level daily case and death data counts in
the Kindom of Saudi Arabia used in this analysis.
(CSV)

## Acknowledgments

The authors would like to acknowledge the many contributors to the COVID Scenario Pipeline model from the Johns Hopkins ID Dynamics COVID-19 Working Group. The authors
are grateful for the overall support provided by Rekha Menon, World Bank Practice Manager,
Health, Nutrition and Population, Middle East and North Africa, and Issam Abousleiman,
World Bank Country Director, Gulf Cooperation Countries. The findings, interpretations,
and conclusions expressed in this work are those of the authors, and do not necessarily reflect
the views of The Ministry of Finance, the Saudi Public Health Authority or the World Bank,
their Boards of Directors, or the governments they represent.

## Author Contributions

**Conceptualization:** Javier Perez-Saez, Elizabeth C. Lee, Justin Lessler.

**Data curation:** Javier Perez-Saez, Reem F. Alsukait.

**Formal analysis:** Javier Perez-Saez.

**Funding acquisition:** Christopher H. Herbst, Justin Lessler.

**Methodology:** Javier Perez-Saez, Justin Lessler.

**Project administration:** Reem F. Alsukait.

**Resources:** Reem F. Alsukait, Christopher H. Herbst.

**Supervision:** Justin Lessler.

**Validation:** Elizabeth C. Lee, Ada Mohammed Alqunaibet, Sami Saeed Almudarra.

**Visualization:** Javier Perez-Saez.

**Writing – original draft:** Javier Perez-Saez, Elizabeth C. Lee, Nikolas I. Wada, Justin Lessler.

**Writing – review & editing:** Javier Perez-Saez, Elizabeth C. Lee, Nikolas I. Wada, Ada Mohammed Alqunaibet, Sami Saeed Almudarra, Reem F. Alsukait, Di Dong, Yi Zhang, Sameh El Saharty, Christopher H. Herbst, Justin Lessler.

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
