## [Decision Letter · Decision Letter 0]

6 Dec 2021

PGPH-D-21-00886

Effect of non-pharmaceutical interventions in the early phase of the COVID-19 epidemic in Saudi Arabia

Dear Dr. Perez-Saez,

Thank you for submitting your manuscript to PLOS Global Public Health. After careful consideration, we feel that it has merit but does not fully meet PLOS Global Public Health’s publication criteria as it currently stands. Therefore, we invite you to submit a revised version of the manuscript that addresses the points raised during the review process.

We look forward to receiving your revised manuscript.

Kind regards,

Thomas P. Van Boeckel

Academic Editor

Journal Requirements:

1. If you have no competing interests to declare, please state "The authors have declared that no competing interests exist". 

2. Please amend your detailed Financial Disclosure statement. This is published with the article, therefore should be completed in full sentences and contain the exact wording you wish to be published.

i). State the initials, alongside each funding source, of each author to receive each grant.

ii). State what role the funders took in the study. If the funders had no role in your study, please state: “The funders had no role in study design, data collection and analysis, decision to publish, or preparation of the manuscript.”

3. Please provide us with a direct link to the base layer of the map used in Figure 1 and ensure this location is also included in the figure legend. 

Please note that, because all PLOS articles are published under a CC BY license (creativecommons.org/licenses/by/4.0/), we cannot publish proprietary maps such as Google Maps, Mapquest or other copyrighted maps. If your map was obtained from a copyrighted source please amend the figure so that the base map used is from an openly available source.

Please note that only the following CC BY licences are compatible with PLOS licence: CC BY 4.0, CC BY 2.0  and CC BY 3.0, meanwhile such licences as CC BY-ND 3.0 and others are not compatible due to additional restrictions. If you are unsure whether you can use a map or not, please do reach out and we will be able to help you. 

The following websites are good examples of where you can source open access or public domain maps:

Additional Editor Comments (if provided):

Dear Authors,

Please find enclosed comment from the referees on your publication " Effect of non-pharmaceutical interventions in the early phase of the COVID-19 epidemic in Saudi Arabia"

In adressing these comment, particular attention you be put on a more extensive description of the methods: including a graphical abstract as suggested by the referee could facilitate this. In addition, greater attention should be placed on the description of the uncertainties associated with fitting model to case data, and comparisons with serology dat as suggested by the referee could help this specific point.

Reviewers' comments:

Reviewer's Responses to Questions

**Comments to the Author**

1. Does this manuscript meet PLOS Global Public Health’s publication criteria? Is the manuscript technically sound, and do the data support the conclusions? The manuscript must describe methodologically and ethically rigorous research with conclusions that are appropriately drawn based on the data presented.

Reviewer #1: Yes

Reviewer #2: Yes

2. Has the statistical analysis been performed appropriately and rigorously?

Reviewer #1: Yes

Reviewer #2: Yes

3. Have the authors made all data underlying the findings in their manuscript fully available (please refer to the Data Availability Statement at the start of the manuscript PDF file)?

Reviewer #1: Yes

Reviewer #2: Yes

4. Is the manuscript presented in an intelligible fashion and written in standard English?

Reviewer #1: Yes

Reviewer #2: Yes

5. Review Comments to the Author

Reviewer #1: This paper presents a mathematical modelling study about the spread of SARS-CoV-2 in Saudi Arabia during the first half year of the epidemic, focusing on the relative reduction in incidence during the different non-pharmaceutical interventions impelemented. The study uses a previously published mathematical model. The paper is well written and the methodology is appropriate. I have the following specific comments:

1) Although the model is already presented in a previous publication, it would be helpful to include a brief description also in this paper (which could be, for example, in the form of a graphical presentation showing the compartments). Although fine to keep the full parameterisation in the appendix, a few key values could also be presented in the main text, mentioning which parameters were fixed and which free. Specific questions:

a) Were the compartments defined based on anything else than infection status and region (e.g., age?)

b) Were the first infections seeded into all regions, or did some regions receive their first cases within the model, by transmission within KSA?

2) Is it correct that you calibrated the model against confirmed cases and deaths (it is mentioned that these data were collected and the model was calibrated, but not explicitly using what indicators). Consider rephrasing. Moreover, fitting a SEIR model against confirmed cases (that depend on a number of factors: true incidence, severity, testing policy, etc) is usually not a very robust choice. What was the assumption regarding the proportion of infections that are recorded as cases?

3) As it seems, from end of May 2020 the restrictions were gradually relaxed. Did anything come in turn? For example, in most of Europe the systematic use of masks, regular testing, effective contact tracing, etc, started to be effectively used only after the initial lockdown.

4) Second sentence of Results (Between 91%...) now slightly complicatedly formulated. What is the meaning? Do these refer to daily numbers of incident cases (e.g. in Al Jafw, 91% of the daily incident case numbers were within the modelled 95% CrI)? Referring to Comment 2, how is the simulated number of daily cases derived from the daily infections?

5) The specificities of KSA as a setting could be discussed more. The differences observed between the regions could be worth discussion. Did the differences you observed in R0 reflect the expectations – for example, was the R0 higher in settings with large and densely populated cities?

6) Figure 4 shows that – against what one would expect – Reff continued to decrease during the relaxation phases. During Phases I and II, this could be an artefact caused by the short duration of these phases; but in some regions, the Reff continued to decrease during Phase III. Do you have any potential explanations for this?

7) I could imagine that there is somewhat lower degree of social mixing in KSA that in many other settings, which limits the applicability of compartmental models that assume heterogeneous mixing within each region. This is a limitation that I think would be worth mentioning and discussing its implications.

Reviewer #2: The manuscript presents a thorough analysis of NPIs in Saudi Ariabia, which adds to the growing literature in support of the impact of NPIs on transmission. While the approach is not novel, its application to a Middle Eastern country is, with the area as a whole significantly less studied than Europe or the Americas. This is a failure of the scientific community and as such I am very encouraged to see this analysis and thoroughly enjoyed reading it.

Overall, I am very happy with the analysis presented and the methods chosen. I have a few comments (somewhere between a minor comment and a major comment), which I think would enhance the manuscript and give more confidence in the findings presented:

1. Seroprevalence data. A recent nationwide blood donor sample of seroprevalence was conducted in Saudi Arabia (Alharbi et al. 2021 July. https://www.sciencedirect.com/science/article/pii/S1876034121000988). While the bias in blood donor sample collection is an open question, it would be very useful to produce plots for each region of the seroprevalence over time inferred from your model fits and compare (if just qualitatively as opposed to formal model fitting) against the observed seroprevalence data. How well does it align and are some regions more aligned with seroprevalence than others? One reason for asking for this is because it would be useful to know if the regions where the model fit to cases incidence is weaker, e.g. Asir, are due to differences in case detection between region. For example, the ratio of deaths to case incidence seems quite consistent between regions but in some it is noticeably higher. Is this because of demographic factors or is it because of different cases detection and thus are the weaker case incident fits are to be expected but we can be confident in the fit to the deaths as the model fit inferred seroprevalence is in agreement with the observed seroprevalence. (Apologies for the longer comment, please feel free to break it down in reply).

2. More information of the model likelihood and model fitting. Currently, it is hard to understand from the SI exactly what model outputs were used for fitting and what distributions were assumed for cases and deaths. E.g. were weekly sums of cases and deaths assumed to be described by a Poisson or Negative Binomial? The SI currency reads as a general overview of a model fitting framework, rather than specifically what the authors did in the analysis, e.g. in the following text:

“The framework allows the user to specify the time unit for a given calibration data; for instance one may calibrate to weekly incident deaths, biweekly confirmed cases, or both.”

3. Please expand on this to describe exactly what model outputs were used to compare to the observed data and what distributions (your p_j terms) these were assumed to take.

In Table S1 please add the region specific estimates for each region as well as the region specific IFRs. Either as a sub table or a new table.

4. In the fitting algorithm, the approach of multiple, shorter chains is fine, however, getting an unbiased estimator of the posterior this way is less valid if the initial conditions for each chain are the same. I assume they are not but it is not stated in the SI. Please add details of how each chain was initialised (the \\Theta_prop terms)

5. Were any priors assumed for the NPI effect sizes as well as the seeding size and data? Were they all assumed to be flat and uninformative? If so please state this.

Other minor comments:

1. All the figures were very clear and it made reading the paper very enjoyable and easy to follow! (just wanted to pass that on)

2. For my own interest, I was wondering how much the between region migration fluxes changed model fitting and behaviour. If these were removed and each region was modelled independently, would different conclusions have been reached?

6. PLOS authors have the option to publish the peer review history of their article (what does this mean?). If published, this will include your full peer review and any attached files.

**Do you want your identity to be public for this peer review?** For information about this choice, including consent withdrawal, please see our Privacy Policy.

Reviewer #1: No

Reviewer #2: No

---

## [Editor Report · Decision Letter 1]

7 Mar 2022

PGPH-D-21-00886R1

Effect of non-pharmaceutical interventions in the early phase of the COVID-19 epidemic in Saudi Arabia

Dear Dr. Perez-Saez,

Thank you for submitting your manuscript to PLOS Global Public Health. After careful consideration, we feel that it has merit but does not fully meet PLOS Global Public Health’s publication criteria as it currently stands. Therefore, we invite you to submit a revised version of the manuscript that addresses the points raised during the review process.

We look forward to receiving your revised manuscript.

Kind regards,

Thomas P. Van Boeckel

Academic Editor

Journal Requirements:

Additional Editor Comments (if provided):

Dear Author,

The reviewers are 'Largely satisfied' with your revisions. However, before formally accepting your manuscript for publication reviewer #2 insisted on the following:

"Related to the analysis of seroprevalence data. The figure showing the comparison of modelled infections is very interesting. The correlation is encouraging that the model is capturing the overall dynamics between regions well. This figure should be included in the SI. Also, when discussing the comparison against seroprevalence in the Discussion, the other explanation for the discrepancy is that deaths have occurred undetected and that the IFR is correct. This alternative explanation must also be listed next to the suggestion that the IFR is incorrect. I would argue strongly that deaths have been undercounted in Saudi Arabia (deaths were missed in all countries at the beginning of the pandemic) and that this contributes to the discrepancy as well, most likely more so than the IFR being overestimated in the current analysis

Once this has been added to the discussion and SI I am very happy to recommend the manuscript for acceptance"

thank you for including this, as suggested.
---

## [Editor Report · Decision Letter 2]

13 Apr 2022

Effect of non-pharmaceutical interventions in the early phase of the COVID-19 epidemic in Saudi Arabia

PGPH-D-21-00886R2

Dear Dr. Perez-Saez,

We are pleased to inform you that your manuscript 'Effect of non-pharmaceutical interventions in the early phase of the COVID-19 epidemic in Saudi Arabia' has been provisionally accepted for publication in PLOS Global Public Health.

Best regards,

Thomas P. Van Boeckel

Academic Editor